# Targeting TRPV4 Channels for Cancer Pain Relief

**DOI:** 10.3390/cancers16091703

**Published:** 2024-04-27

**Authors:** Caren Tatiane de David Antoniazzi, Náthaly Andrighetto Ruviaro, Diulle Spat Peres, Patrícia Rodrigues, Fernanda Tibolla Viero, Gabriela Trevisan

**Affiliations:** 1Graduate Program in Pharmacology, Federal University of Santa Maria (UFSM), Santa Maria 97105-900, Brazil; caren.antoniazzi@ufsm.br (C.T.d.D.A.); diulle.peres@acad.ufsm.br (D.S.P.); patricia.rodrigues@acad.ufsm.br (P.R.); fernanda.viero@acad.ufsm.br (F.T.V.); 2Graduate Program in Toxicological Biochemistry, Federal University of Santa Maria (UFSM), Santa Maria 97105-900, Brazil; nathaly.ruviaro@acad.ufsm.br

**Keywords:** nociception, osmotransducer, tumor microenvironment, CIBP, CIPN, TRP channels

## Abstract

**Simple Summary:**

Cancer stands as a major public health concern worldwide, and a significant proportion of survivors are affected by recurrent and usually unrelieved cancer pain. The participation of ion channels in cancer pain’s initiation and maintenance is well-described; however, the evidence on the contribution of transient receptor potential vanilloid 4 channels is scarce. The limited studies published to date encountered enhanced TRPV4 expression in cancer-induced bone pain (CIBP), perineural, and orofacial cancer models, increasing pain perception. These outcomes support the hypothesis that the modulation of TRPV4 channels can be targeted as a novel therapeutic approach for treating difficult pain. This review summarizes the role of TRPV4 in cancer etiology and cancer-induced pain mechanisms, as well as the current status of the clinical research.

**Abstract:**

Despite the unique and complex nature of cancer pain, the activation of different ion channels can be related to the initiation and maintenance of pain. The transient receptor potential vanilloid 4 (TRPV4) is a cation channel broadly expressed in sensory afferent neurons. This channel is activated by multiple stimuli to mediate pain perception associated with inflammatory and neuropathic pain. Here, we focused on summarizing the role of TRPV4 in cancer etiology and cancer-induced pain mechanisms. Many studies revealed that the administration of a TRPV4 antagonist and TRPV4 knockdown diminishes nociception in chemotherapy-induced peripheral neuropathy (CIPN). Although the evidence on TRPV4 channels’ involvement in cancer pain is scarce, the expression of these receptors was reportedly enhanced in cancer-induced bone pain (CIBP), perineural, and orofacial cancer models following the inoculation of tumor cells to the bone marrow cavity, sciatic nerve, and tongue, respectively. Effective pain management is a continuous problem for patients diagnosed with cancer, and current guidelines fail to address a mechanism-based treatment. Therefore, examining new molecules with potential antinociceptive properties targeting TRPV4 modulation would be interesting. Identifying such agents could lead to the development of treatment strategies with improved pain-relieving effects and fewer adverse effects than the currently available analgesics.

## 1. Introduction

Despite the advances in modern medicine to prevent and diagnose malignant tumors and the improvement in treatment strategies to reduce the prevalence and mortality of oncological patients, cancer remains a public health concern worldwide [1]. According to the latest update from the International Agency for Research on Cancer (IARC), there were 20 million new cases of cancer and 9.7 million cancer-related deaths reported globally in 2022 [2], with cancer remaining one of the leading causes of morbidity and mortality [3]. However, the mortality rates have been declining in recent years because of the effectiveness of novel therapies developed for cancer treatment. This increase in oncological patient survival also means that more patients are likely to live with long-lasting symptoms arising or persisting beyond the conclusion of therapy [4].

A significant proportion of survivors are affected by complications and sequelae, such as chronic pain, a secondary symptom linked to the underlying disease [5]. Cancer-related pain is recurrent and one of the most invalidating symptoms experienced by oncological patients, seriously interfering with their daily functioning, motivation, and personal relations with family and friends, reducing their overall comfort and quality of life [6,7]. Many patients will endure longer periods of cancer-related severe pain, enough to require opioid therapy [1,8,9], and more than one-third of those in disease remission will continue to report unrelieved pain [1,3]. Long-term cancer pain is highly prevalent during and up to three months after curative cancer therapies, affecting 40% of patients [10]. Also, a recent analysis revealed that pain is prevalent for more than 50% the patients undergoing cancer treatment, and this ratio is even higher for patients with advanced metastatic disease [1].

Unalleviated pain has emerged as a growing problem, since pain management related to the adverse effects of cancer and its treatment have remained mostly the same over the last few decades [1,11]. The current guidelines to assist with cancer pain relief were first published in the 1980s by the World Health Organization (WHO) and consist of a stepwise method to guide healthcare providers, in which the severity of pain determines the adequate choice of analgesic [12,13]. Over the years, the successive adjustments made to these standard procedures allowed for their continued use as a valid strategy to control cancer-related pain by providing a simple, palliative approach to ease pain-related morbidity in more than 70% of patients [14,15]. These continuous updates to the guideline allowed for significant advances in cancer pain management by including previous pain assessments, distinguishing pain according to the cancer stage (e.g., acute pain after treatment, pain in advanced or palliative cancer, pain in cancer survivors), and highlighting the importance of non-pharmacological treatments [15,16].

According to the WHO guidelines, the treatment starts with over-the-counter analgesics, such as non-steroidal anti-inflammatory drugs (NSAIDs) and acetaminophen, to manage mild pain. A second stage escalates the treatment to medications considered weak opioids (tramadol and codeine) to manage mild to moderate pain. Potent opioids such as morphine, buprenorphine, fentanyl, and oxycodone are recruited as a third step and used to control severe pain [12,13,14]. Although opioids still stand as the primary choice for the pharmacological treatment of cancer pain, weak opioids have a minimal impact on treating cancer-related pain, and a review of strong opioids did not find definitive evidence on the efficiency of those medications in contributing to adequate pain relief [17]. Thus, adjuvant therapies should be considered at any step of the treatment for improved pain control, particularly neuropathic pain. This complementary care may include anticonvulsants (gabapentin and pregabalin), serotonin–norepinephrine reuptake inhibitors (duloxetine), tricyclic antidepressants (amitriptyline and desipramine), local anesthetics (lidocaine), or ion channel activators (capsaicin and menthol, agonists of transient receptor potential vanilloid 1 [TRPV1] and melastatin 8 [TRPM8], respectively) [14,18,19].

More recently, cannabinoids have been given increased attention due to their effects on reducing inflammation and cancer-related pain, providing evidence that adjuvant therapy with these compounds may lead to a reduction in opioid doses and opioid-related mortality [20,21,22]. Another alternative relies on the use of neurotoxins, such as resiniferatoxin, tetrodotoxin, and botulinum toxin. Compared to standard analgesics, these molecules are more potent for treating moderate to severe and refractory-cancer-related pain, in addition to their non-addictive and long-lasting therapeutic properties after the last day of treatment [23,24,25,26,27]. Moreover, despite there being no clear recommendations for this, ketamine, an N-methyl-D-aspartate (NMDA) antagonist, has also been included as an adjuvant for cancer pain treatment. It may be helpful for specific subgroups of patients, such as those affected by central sensitization. However, the studies on this are controversial, and convincing clinical evidence is scarce [28,29].

A final, fourth step can be considered when healthcare providers detect a lack of proper analgesia or avoid high doses of opioids, which is closely related to the occurrence of intense adverse events [16,30]. These integrative therapeutic options suggest the need for assorted non-pharmacological and interventional practices for treating severe pain [31] and embrace techniques like acupuncture and physiotherapy, neuromodulation, nerve block, ablative procedures, and epidural/intrathecal analgesics [12,13,16,30]. Based on the assumption that early oncological pain management reduces suffering and improves patients’ quality of life, there are indications that the primary use of interventional procedures could be more beneficial if provided sooner in the disease course to control symptoms and prevent opioid dose escalation, rather than offering these options only when pain is considered refractory to standard pharmacological treatment [13,32].

Cancer pain is an intricate condition, driven by inflammatory, neuropathic, and cancer-specific mechanisms. This combination of factors can compromise the proper management of pain following cancer and its treatment, reflecting the high prevalence of cancer-related pain among this population [16,30]. Regardless of the source, uncontrolled pain hinders patients’ mental and physical well-being, affecting their daily routine [16,33].

## 2. Cancer Pain Mechanisms

The development of severe and persistent pain associated with cancer encompasses complicated and poorly understood mechanisms, including successive changes in cellular, tissue, and systemic levels [11]. Countless factors can cause these alterations, but substances secreted by tumor cells stimulate tumor-infiltrating sensory nerve endings, leading to nociceptor sensitization [34]. Meanwhile, cancer cell proliferation and metastatic infiltration can compress, permeate, and destroy surrounding tissues, affecting adjacent sensory fibers and leading to neuropathic alterations [35,36], as observed in cancer-induced bone pain (CIBP). Characterized by the migration of tumorigenic cells to the bones, CIBP is a debilitating symptom and the most common cause of cancer-related pain. This unique condition is regulated through nociceptive, neuropathic, and inflammatory processes. CIBP usually emerges from tumors and associated stromal cells releasing several mediators (cytokines, chemokines, and prostaglandins), cancer growth compressing the spinal cord and bone neuronal terminals, and tumor-bearing bone fractures. Consequently, the combination of all these aspects leads to pain, hypercalcemia, anemia, increased susceptibility to infections, spinal instability, and decreased mobility [37,38,39].

Cancer treatments provided through surgical procedures, chemotherapy, and radiotherapy likewise can cause further damage and contribute to the exacerbation of chronic painful states [11,38]. Chemotherapy-induced peripheral neuropathy (CIPN) arises from antitumor agents’ neurotoxicity impairing peripheral sensory, motor, and autonomic nerves. Besides the painful sensation, this damage also triggers paresthesia, tingling, and burning sensations in the hands and feet, which characterize CIPN [40,41]. CIPN can be challenging to treat and often worsens the patient’s suffering, as the clinical guidance underlines the absence of preventative strategies and the scarcity of options for symptom management [40,42].

In addition, cancer induces plasticity in the peripheral and central nervous system, and these interactions between the tumor, primary afferent nociceptor, and immune system contribute to the unpredictable aspects of cancer’s phenotypic and genomic heterogeneity, which is why managing this type of pain can be challenging [9]. The sensory afferent terminals express a wide range of receptors, like transient receptor potential ankyrin 1 (TRPA1), members of the vanilloid subfamily (mainly TRPV1 and TRPV4), and some members of the melastatin group (TRPM2, TRPM3, and TRPM8), which were all described to be involved in pain perception [43].

The activation of non-TRP channels, such as Piezo (involved in the conversion of mechanical stimuli into electrical signals) [44], P2X receptors (activated by extracellular adenosine triphosphate [ATP]) [45], acid-sensing ion channels (ASICs, sensitive to extracellular acidification as detected in the tumor microenvironment) [46], and cannabinoid receptors (CB1, expressed in the central nervous system and nerve terminals, and CB2 in the immune system), also acts to control the transduction of pain from nociceptive stimuli, the production of inflammatory cytokines by the nerves and immune cells, and the activation of sensory neurons [39].

Moreover, both pain initiation and the maintenance of cancer pain were associated with an enlarged expression of nerve growth factor (NGF) and brain-derived neurotrophic factor (BDNF), and the increased release of tryptase and ATP [8,9,18,47]. Other algogenic substances, such as tumor-producing cytokines and tumor necrosis factors (TNF), also release nociceptive mediators because of the proteolytic activity induced by tissue damage, favoring nociceptor sensitization, which finally contributes to the sustenance of cancer-related pain [48]. In addition to the intense and persistent pain, emotions (distress, anxiety, and depression), cognition, and social context play an important role in cancer pain, impacting the patient’s ability to control pain [49]. Other common symptoms, such as fatigue, sleep deprivation, and appetite disorders, might eventually interfere with pain processing [8,50].

In many instances, patients may perceive pain through a mixed pathophysiology. Thus, to improve the relief of patients’ cancer-related pain, it is critical to specify the predominant type of pain and its cause (tumor growth, cancer metastasis, treatment-related, or other comorbidities), and it might require the pharmacologic and non-pharmacologic antagonism of mechanisms in the neuro-sensory system to cancer proliferation [8,51]. The transition from a symptom-based approach to a more mechanistic-based classification is relatively recent. This considers the differential classification, well-defined by the International Association for the Study of Pain (IASP) as nociceptive, neuropathic, and nociplastic pain, which is extensively used in pain research [4,49,51].

Despite the recent advances in cancer pharmacotherapies, traditional pain management still relies on opioids and standard first- and second-line analgesics, which are usually indicated for neuropathic pain conditions [9,30]. Opioid drugs, cannabinoids, and serotonergic agonists can activate the mechanism of the central modulation of pain, providing neuronal activation in brain areas that inhibit or control pain, which are unquestionably critical pharmacological targets for treating chronic pain conditions such as those involving cancer-related pain [18,52,53]. This type of pain continues to be overlooked by healthcare providers, with patients frequently not being asked or not receiving proper education about their pain [54].

Because of the poor understanding of the mechanisms involved in its etiology, cancer pain remains under-investigated, and in addition to the lack of selective targets for developing novel, effective, and safe drugs, its management is often suboptimal, with patients undergoing unsatisfactory treatments [4,54]. Due to their central involvement as mediators in the mechanisms of painful stimulus transmission between the central and peripheral nervous system, sensory afferent fibers are attractive candidates to alleviate this symptom. In this context, the relevance of TRP channels stands out, as they represent the most important ion channel family in detecting and transmitting noxious stimuli. As TRP channels are widely expressed by nociceptive neurons and distributed along the fibers and somas [55,56], they arise as promising candidates for the development of novel pharmacotherapies suitable for treating a variety of pain conditions, which include cancer-related pain.

## 3. Transient Receptor Potential (TRP) Channels

Transient receptor potential channels form a group of non-selective, cation-permeable channels that are extensively expressed in many tissues and cell types. These channels are critical for many physiological functions, including detecting numerous stimuli and maintaining the ion balance within cells. Since their initial characterization, the members of mammalian proteins identified as TRP channels have significantly increased and now encompass a superfamily of 28 isoforms, sorted into six subfamilies based on the similarity of their genetic sequences. The subfamilies include ankyrin (TRPA1), canonically consisting of TRPC1 to TRPC7; melastatin, which comprises TRPM1 to TRPM8; mucolipin, with TRPML1 to TRPML3, polycystin, containing TRPP1 to TRPP3; and vanilloid, which includes TRPV1 to TRPV6 [57].

These receptors are crucial in detecting and transducing sensory stimuli into electrical signals. They are mainly regulated by changes in messenger molecules, mechanical forces, chemical compounds (endogenous and exogenous), pH, and osmotic pressure [19,58]. Oxidative stress by-products, including hydrogen peroxide (H_2_O_2_), can also trigger TRP isoforms (TRPA1, TRPV1, and TRPV4). Moreover, several TRP channels behave as thermoreceptors in the peripheral sensory nerves, such as TRPA1 and TRPM8, fulfilling an essential role in the detection and response to noxious cold (≤17 °C) or cold temperatures (≤28 °C) respectively, TRPV1 and TRPV2, which are sensitive to noxious heat (≥43 °C and ≥52 °C, respectively), and TRPV4, which is activated by warm temperatures in the range of 34–38 °C [19,57,59].

TRP channels can be involved in taste, thermal, and mechano-sensations, but many of these members are fundamental proteins in pain perception and the integration of nociceptive signals. The activation of TRP sensors increases the intracellular concentration of cations following cellular membrane depolarization, resulting in protective responses such as pain and local inflammation [56,58,60]. Moreover, the abnormal functioning of these channels (‘TRP channelopathy’) can be linked to several pathological conditions like neurodegenerative disorders, neuropathic pain, and cardiac and renal diseases, and these ion channels were among the first to be consistently associated with cancer [60,61,62]. TRPM8 was first detected as a protein overexpressed in prostate cancer, and its overexpression was associated with tumor-characteristic events such as cell migration and invasion. Also, the role played by TRPA1 in the development of nociception was well described in murine models of melanoma, breast cancer, and metastatic bone cancer pain [63,64,65].

Similarly, different TRPV isoforms have been involved in the development and progression of several types of cancers, including oral, bone, colon, prostate, pancreatic, and skin. Altogether, the de-regulated expression of these channels has emerged as a crucial factor that influences multiple hallmarks of cancer biology, covering critical processes such as tumor cell proliferation, migration, invasion, angiogenesis, and resistance to therapeutic measures [62,66]. Recent reports have proposed the existence of an intricate connection between TRPV4 activation and the tumor dynamic microenvironment, suggesting that during this complex process, TRPV4 channels sensitized by different stimuli, including matrix stiffness, are rapidly activated and became critical for the arrangement of the metastatic cascade in a plethora of tumors, such as breast, endometrial, gastric, and colon cancer, as well as in the induction of the migratory phenotype in melanoma cells [67,68]. The emerging recognition that the aberrant activity of TRPV channels interferes with the context of cancer pathophysiology highlights them as potential candidates for pharmacological intervention. Regarding this, TRPV4 has been increasingly recognized, especially its possible role in suppressing cancer cell metastasis. Modulating the activity of these channels could lead not only to the inhibition of tumor cells’ growth but also to the increased efficacy of existing therapies on TRPV4-overexpressing cells [19,61,66,67].

In the current review, we discuss the involvement of TRPV4 channels in the intricate process of cancer-induced pain. Here, we focused on TRPV4’s contribution to the pain sensation during cancer development and its manifestation. Although many studies have already addressed the role of other TRP channels, including TRPA1 and TRPV1, in the process of cancer pain development [69,70], our review mainly focused on TRPV4’s action, addressing the uniqueness of this protein.

### 3.1. Transient Receptor Potential Vanilloid 4 in Cancer Etiology

The topological structure of TRPV4 is an ingenious arrangement of six transmembrane domains designated as S1–S6. A loop between segments S5–S6 forms the channel pore, facilitating the ionic flow necessary for the cell’s activity. The protein’s N- and C-termini regions are localized in the cell membrane, while cysteine residues are found in extra- and intracellular environments. The unique feature of S1–S4 domain packing against the S5–S6 transmembrane domains differentiates TRPV4 from the other TRPV isoforms, a topological arrangement that gives rise to its unique functional properties [59,67]. TRPV4 is an ionotropic receptor acting as a polymodal sensor of non-selective cation channels. Initially identified as an osmotransducer, TRPV4 was recognized as a molecular sensor of mechanical stimuli, including pressure and shear stress, particularly in peripheral nerves and muscular tissues [71,72,73]. Recent studies have expanded our understanding of its molecular mechanisms, identifying TRPV4’s role as a mechanosensory receptor. The expression of TRPV4 in trigeminal (TG) and dorsal root ganglion (DRG) sensory neurons suggests its potential involvement in pain signaling and perception [71,73,74,75,76]. Further research elucidating TRPV4’s versatile activity in cell physiology and sensory perception will unveil novel insights into this target as a potential therapeutic candidate for many pathologies, including pain disorders.

In addition to hypoosmotic conditions and non-noxious heat (34–38 °C), TRPV4 responds to many other activators. These include cellular swelling, acidic environments (pH 5.0), H_2_O_2_ exposure, and ultraviolet B-rays (UVB) [67,77,78,79,80]. Furthermore, the involvement of TRPV4 in nociception is modified by multiple pro-nociceptive factors, including the prostanoid PGE_2_, the activation of protease-activated receptors PAR-2 signaling, inflammatory mediators, and nerve injury, all of which have been confirmed to increase TRPV4-dependent pain signaling in several physiological systems [73,74,78,81,82,83]. TRPV4’s responsiveness to these agents suggests a combination of exogenous and endogenous chemical substances that alter the activity of this channel. Among the endogenous agonists that bind and activate TRPV4 are anandamide, arachidonic acid and its metabolite 5′,6′-epoxyeicosatrienoic acid (5′,6′-EET), and dimethylallyl pyrophosphate [77,80,84].

On the other hand, exogenous factors capable of TRPV4 activation included a broad spectrum of compounds, like 4α-phorbol 12,13-didecanoate (4α-PDD) and bisandrographolide A (both molecules derived from plants), ω-3 polyunsaturated fatty acids, and synthetic agents like GSK1016790A [77,85]. Moreover, while other compounds, including ruthenium red, gadolinium, and lanthanum, have long been employed to inhibit TRPV4, they lack specificity, making them impractical for experimental purposes [86,87]. Nevertheless, some progress has been made in recent years to develop selective TRPV4 antagonists in an attempt to alleviate nociceptive responses. Preclinical evidence has identified several compounds, such as HC-067047, RN-1734, and GSK2193874, that block TRPV4 and significantly lower pain sensitivity. However, it is worth mentioning that the translation of these findings into clinical practice has been met with mixed results. The only TRPV4 antagonist to enter clinical trials, GSK2798745, has demonstrated limited efficacy, suggesting that TRPV4 channel regulation for analgesic purposes in patients may be particularly difficult [87,88,89].

TRPV4 is detectable in a wide range of mammalian cells, such as immune cells (e.g., macrophages, neutrophils, and dendritic cells), sensory and cortical pyramidal neurons, glial cells, keratinocytes, skeletal muscle fibers, and tissues like the spinal cord, thalamus, and cerebellum basal nuclei. Furthermore, these channels were functionally related to cell proliferation, differentiation, apoptosis, migration, and other mechanisms, making them essential to many physiological processes (Figure 1A). On the other hand, TRPV4 appears to be upregulated in a variety of pathological conditions, being associated with inflammatory diseases affecting the central and peripheral nervous system, such as osteoarthritis, atherosclerosis, cancer pain, and neuropathies [62,73,90].

As TRP channels’ involvement in carcinogenesis has been increasingly recognized, research on these channels as potential drug targets in many cancer conditions has recently attracted much attention. Previous studies reported that the anomalous expression or function alterations of ion channels might drive normal cells to transform into tumorigenic derivatives, although this process involves a complex network [91,92]. Also, some tumor cell lines display an unusual ion channel expression, which interferes with the typical patterns of intracellular local Ca^2+^ distribution, resulting in the imbalance of multiple downstream effectors dependent on Ca^2+^ homeostasis [91]. This altered ion transport through TRPV4 channels is highly relevant to essential aspects of human cancer pathophysiology. It strongly contributes to cancer cells’ aggravated proliferation, survival, and invasion capacity (Figure 1B), as evidenced by multiple correlated studies [62,68,91,93,94]. These studies demonstrated that TRPV4 expression also varies among different tumors, indicating the dynamics of its overall involvement in tumorigenesis and disease progression and correlating this atypical regulation of TRPV4 with the relentless progression and invasive nature of tumors.

The complexity of TRPV4 extends beyond its expression patterns, with the channel actively participating in the regulation of cellular functioning, which is associated with cancer progression. Specifically, TRPV4 contributes to the dysregulation of cell proliferation, promoting uncontrolled neoplastic growth while actively engaging in the abnormal differentiation patterns observed in malignant cells (Figure 1B). Moreover, emerging evidence suggests that TRPV4 plays a crucial role in the autophagy phenomenon, complicating the cancer landscape and impacting the oncological patient prognostics [68,91,92].

Evidence from previous studies confirmed that TRPV4 channels regulate Ca^2+^ influx and release in human hepatoblastoma cells; meanwhile, the hepatocyte growth factor/scatter factor activates TRPV4 and TRPV1 channels, gradually amplifying the signaling and leading to cell mortality [95]. The pharmacological activation of TRPV4 promotes cell death via oncosis and apoptosis in breast cancer cells overexpressing TRPV4 [96], and another study found that these channels are required for breast cancer metastasis and transendothelial migration but had no effect on cell growth/proliferation [97]. Moreover, the reduced expression of TRPV4 in tumor endothelial cells caused an increase in proliferation, confirming that these receptors may regulate abnormal tumor angiogenesis [98,99]. Among multiple cancers, skin cutaneous melanoma possesses a higher abundance of TRPV4, and in this type of cancer, the acquired results indicated that TRPV4 promotes metastasis through cell motility regulation [68]. The selective reduction in TRPV4 expression in human non-melanoma skin cancer characterizes TRPV4 as a possible early biomarker of skin carcinogenesis [100]. In models investigating bladder cancer tissue and para-carcinoma tissue, the variations in TRPV4 expression were not evident [101,102], but TRPV4 was able to detect mechanical and chemical stimuli, induce calcium influx, and promote ATP release [103]. Regardless of the many studies conducted to unveil TRPV4’s role in cancers, the channel expression variability in different tumor tissues and the underlying mechanisms are still largely undetermined [67]. Although developing TRPV4-based antitumor drugs continues to be challenging, therapeutic benefits could emerge from TRPV4’s potential inhibitory effect on tumor onset and progression, enhancing the prognostic index [19,67,91,92].

TRPV4’s participation has also been investigated in different preclinical models of pain and nociception induced by inflammatory and neuropathic pain [73,83,104,105,106,107], and this receptor was implicated as a contributor to the etiology of chemotherapy-induced peripheral neuropathy [59,106].

### 3.2. The Role of TRPV4 in the Chemotherapy-Induced Peripheral Neuropathy (CIPN)

Chemotherapy-induced peripheral neuropathy (CIPN) is a primary dose-limiting adverse effect and a frequent complication affecting a significant proportion of patients undergoing chemotherapy [40,41]. Although the neurotoxicity levels of antineoplastic drugs vary, many agents have been associated with CIPN, including taxanes (e.g., paclitaxel and docetaxel), immunomodulatory agents (e.g., thalidomide), platinum derivatives (cisplatin and oxaliplatin), vinca alkaloids (vincristine), and proteasome inhibitors (bortezomib) [41,108].

The onset and severity of symptoms are also drug-specific and usually reliant on the dosage, frequency, route of administration, and therapy duration [40,109]. Most signs and symptoms of CIPN arise from damage to the longest axons in distal nerves. Typical CIPN manifestation follows a “glove and stocking” anatomical distribution, simultaneously affecting patients’ hands and feet [42,108]. Symptom progression is distal-to-proximal, leading to acral pain, sensory loss (numbness, tingling, burning, paresthesia/dysesthesia, and/or hyperalgesia/allodynia), and occasionally sensory ataxia (loss of limb strength, impaired proprioception, and declined perception of external touch and vibration) can be present [42,108]. Taken together, sensory disturbances and neuropathic pain profoundly impact the quality of life of cancer survivors, affecting daily functioning and the capacity to work and leading to unemployment and loss of working time [40].

Intricate and multi-factorial mechanisms have been described to be involved in CIPN, depending on the antineoplastic agent of choice [61,108,109]. The most extensively accepted mechanism is related to a “dying back” process with the axonal degeneration of sensory neurons, resulting in the loss of intra-epidermal nerve endings. The impaired axonal transport leads to a deficiency in the supply of essential nutrients responsible for synapse formation and maintenance [61,108,109]. Suggested mechanisms, such as irreversible cell injury, oxidative status imbalance, impairments in ATP production secondary to mitochondrial damage and dysfunction, the abnormal excitability of pain fibers (Aδ and C fibers), and neuroinflammation (the activation of macrophages in both the dorsal root ganglion and peripheral nerve, and the activation of microglia cells within the spinal cord) are also considered to translate chemotherapy-induced neurotoxicity into CIPN [40,61,109].

Another prominent mechanism implicated in CIPN relies on the altered expression and functional impairment of voltage-gated ions (Na_v_^+^, K_v_^+^, Ca_v_^2+^) and TRP channels [61,110]. The dysregulation of calcium homeostasis and signaling influences the activation of voltage-gated ion channels, leading to neuronal hyperexcitability, neurotransmitter release, and the gene expression of neuronal cells [61,108,110]. These mechanisms have been targeted for therapeutic intervention, and preclinical models have investigated the involvement of the TRPV4 channel in neuropathic pain, a characteristic symptom described in CIPN etiology (Figure 2).

Most studies investigating TRPV4’s involvement in CIPN-induced nociception have reported paclitaxel, thalidomide, and vincristine as causing neural injury, leading to painful peripheral neuropathy (Figure 2A,B). These antineoplastic drugs were able to intensify TRPV4 functionality, mainly due to TRPV4’s involvement in the mechanical hyperalgesia and hypotonic-induced nociception evoked by these agents [78,81,106,111,112,113]. TRPV4 channels contribute to the development of nociceptive behaviors participating in a molecular complex that includes the activation of the α2β1 integrin (implicated in the transduction of osmotic stimuli and mechanical stretch) and at least one member of the Src tyrosine kinase family (involved in ion channels’ modulation) signaling pathways (Figure 2C,D) [78,112].

In preclinical experiments, paclitaxel was found to increase TNF-α expression, leading to the upregulation of TRPA1 and TRPV4 channels in dorsal root ganglion (DRG) neurons (Figure 2E–G), and blocking TNF-α signaling alleviated pain-like symptoms and prevented TRPA1/TRPV4 upregulation [114]. Also, paclitaxel toxicity increased the TRPV4 channel expression and TRPV4-dependent calcium fluxes in neuronal cells, and these adverse effects were reduced by treatment with lithium, which is clinically used to prevent CIPN [106]. It was reported that TRPV4 is often co-expressed with TRPC1 and TRPC6 in DRG neurons, which may indicate that these channels work in consonance. This connection has been hypothesized to underly the development of mechanical hyperalgesia and, ultimately, the sensitization of primary afferent nociceptors elicited by chemotherapeutic agents such as paclitaxel or cisplatin [111]. The co-expression of these channels suggests a general pain mechanism in CIPN, since TRPV4, TRPC1, and TRPC6 might collectively mediate pain sensitivity enhancements. In addition, the administration of paclitaxel is shown to trigger the release of mast cell tryptase, an enzyme that activates PAR-2. The latter subsequently activates protein kinases A and C, causing the sensitization of TRPV4 channels (Figure 2H), and as a result of these signaling pathways, mechanical and thermal hypersensitivity are observed, which also indicates the involvement of TRPV4 in the development of CIPN [81].

The studies carried out in murine models have proved the efficacy of the genetic deletion or knockdown of *Trpv4* gene. According to the findings, the absence of *Trpv4* significantly and consistently diminishes nociceptive responses in several preclinical models. The intrathecal administration of antisense oligodeoxynucleotides targeting TRPV4 receptors in nociceptive sensory neurons results in the downregulation of ion channel expression. This approach not only reversed the chemotherapy-induced mechanical allodynia evoked by paclitaxel and vincristine but also altered the severity of hypotonic hyperalgesia in animal models. Moreover, the evidence from *Trpv4* wild-type mice subjected to chemotherapies showed enhanced nociceptive responses. This means that these responses were considerably more robust than those in *Trpv4* knockout animals [78,112,113]. In addition, the blockage of Ca^2+^ influx through TRPV4 channels with antagonist HC-067047 prevented cell death in cultured neurons, averted the electrophysiological and behavioral changes associated with paclitaxel-induced neuropathy [115], and partially attenuated the mechanical allodynia and oxidative stress evoked by paclitaxel in preclinical models [79].

Another study demonstrated that kinins sensitize TRPV4 channels to maintain mechanical hyperalgesia and increase nociception in response to hypotonicity induced by paclitaxel, which TRPV4 or kinin receptor antagonists notably alleviated [116]. Thalidomide- and paclitaxel-induced neuropathies activate TRPA1 and TRPV4 channels through the excessive production of oxidative stress by-products like H_2_O_2_ (Figure 2I), and the co-administration of TRPA1 (HC-030031) and TRPV4 (HC-067047) antagonists contributed to the complete ablation of allodynia and oxidative stress [79,113]. RN-1734, another TRPV4 antagonist, also prevented the sensitization of TRPV4 and attenuated paclitaxel-induced neuropathic pain [81].

In summary, these results highlight the crucial role played by TRPV4 in sustaining the painful status affecting a significant portion of patients with CIPN [43,117]. Despite the high incidence, there has been no efficacious therapeutical option approved to alleviate CIPN symptoms to date, translating to an impaired patient quality of life. Pharmacological agents and non-pharmacological interventions have been recommended or discouraged based on the available evidence of their potential to prevent neuropathy [118,119]. Thus, the alleviation of CIPN symptoms via TRP channels is now being investigated in multiple studies. Current data collections regarding human subjects and experimental animals imply the involvement of at least five thermo-TRP channels—TRPA1, TRPM8, TRPV1, TRPV2, and TRPV4—in CIPN pathogenesis and progression. Recent discoveries of TRP’s channel activity and modulation led to the development of several channel modulators, which have reached clinical trials. As evidenced by several recent reviews, these modulators have shown promising characteristics to manage a variety of pain types, including inflammatory, neuropathic, and visceral pain. Regardless of the clinical relevance, however, the precise mechanism of action leading to CIPN symptom amelioration remains uncertain [66,117]. Therefore, research in this field continues to unfold, and further elucidation of TRP channels’ molecular function and their contribution to CIPN pathophysiology is warranted. Given the extensive level of TRPV4 channel expression in sensory and immune cells and its broad participation in physiological processes and diseases [59,61,117], the potential relevance of targeting TRPV4 inhibitors as therapeutical options to treat painful neuropathy associated with antineoplastic drugs is suggested [19,87].

### 3.3. TRPV4 Participation in Cancer-Induced Pain

Acute or chronic pain is recognized as the most recurrent and distressing symptom associated with cancer and may exist during cancer care and beyond [8,48]. The challenges in managing cancer-related pain can be exhausting as the mechanisms driving cancer pain diverge from those responsible for inflammatory and neuropathic pain. As previously described, cancer pain can be influenced by many factors, either directly associated with tumor development and progression or with the diagnostic and therapeutic procedures and interventions [9,120]. However, the prevailing theory to elucidate cancer pain hypothesizes that tumors generate and release signaling molecules, which, besides regulating metastasis and tumor growth, also sensitize and activate primary afferent nociceptors in the tumor microenvironment [8,121]. Some molecules have been reported to directly or indirectly modulate TRPV4 channel activity, including TNF-α, nerve growth factors, prostaglandins, and bradykinin [77,114,122]. Combined with the acidification detected in the tumor microenvironment, these mediators can modulate TRP channels activity via the regulation of intracellular signaling pathways (Figure 2J), resulting in the peripheral sensitization described by oncological patients, in which mild noxious sensory stimuli are perceived as hyperalgesia and allodynia [121,122].

TRPV4 mediates the nociceptive responses triggered by hypotonic stimuli, low pH, pressure, oxidative stress by-products and, likely to other TRP channels, significantly impacts the mechanotransduction of pain related to peripheral neuropathies through its prominent expression in peripheral sensory neurons [77,86,122]. As mentioned before, these channels are primarily expressed in DRG and TG sensory neurons, making a crucial contribution to the pain caused by inflammation [74,75] and controlling mechanical and thermal hyperalgesia in neuropathic pain [113]. Moreover, TRPV4 activation in dural afferents leads to hind paw and facial allodynia in mice, while mechanical hyperalgesia is triggered by PAR-2 activation in primary afferent neurons expressing TRPV4. These ion channels are also expressed in immune cells, where their activation results in Ca^2+^ influx [66,82,122]. Considering this, it is pertinent to investigate the processes through which TRPV4 is involved in cancer pain.

It is important to mention that, in practice, the impact of cancer metastasis on the skeletal system results in a spectrum of devastating consequences, including compression of the spinal cord and other nervous structures, bone fractures, immobilization, and, eventually, enhanced mortality, with patients forced to endure recurrent episodes of cancer-induced bone pain (CIBP) [37,38,39]. Recent findings provide further evidence of the role of TRPV4 channels in the genesis of bone metastases and the continuance of the pain in CIBP [123,124,125]. The activation of TRPV4 in osteoclasts causes an increase in bone reabsorption, promoting the formation and progression of bone metastases [39,124]. Similarly, the activation of TRPV4 channels expressed on the sensory neurons that innervate the bone stimulates the release of neurotransmitters signaling pain. This upregulation in TRPV4-mediated signaling pathways is critical for pain generation and persistence [39,123,125].

Although TRPV4 represents a compelling pharmacological target, only a few preclinical studies have investigated its involvement in cancer-related pain, as summarized in Table 1.

As TRPV4’s role in neuropathic pain was already established [83], this was the first study to confirm that TRPV4 is also involved in cancer pain. The CIBP model was designed to investigate the involvement and upregulation of these channels in DRG neurons and whether treatment with an herbal compound named Long–Teng–Tong–Luo (LTTL) would decrease receptor overexpression to inhibit pain. The topical treatment with LTTL started on the day after cancer cells’ implantation in the medullary cavity of the rat’s tibia. The boosted expression of TRPA1-, TRPV1-, and TRPV4-immunoreactive cells in ipsilateral L4–5 DRG due to bone cancer proliferation was inhibited by LTTL gel application. LTTL downregulation of TRP channel expression alleviated bone-cancer-induced mechanical allodynia and thermal hyperalgesia, suggesting a correlation with its inhibitory effect on CIBP. Cancer-bearing rats showed an enhanced expression of IL-17A in the spinal astrocytes but not in microglia and neurons. Many pro-inflammatory spinal cytokines, such as TNF-α, interleukin-1β (IL-1β), and IL-17, act as neuromodulators, regulating pain via neuron–glial or glia–glia interactions and are reported to be involved in the pathogenesis of chronic pain and bone cancer pain [126]. LTTL treatment lowered IL-17A overexpression while the intrathecal administration of IL-17A antibodies mitigated bone-cancer–evoked nociception in a dose-dependent manner, revealing that astrocyte-produced IL-17A might endorse cancer-related pain. Also, the intrathecal administration of the TRPA1 antagonist (HC-030031) and non-selective antagonists for TRPV1 (iodoresiniferatoxin) and TRPV4 (gadolinium) downregulated bone-cancer-enhanced IL-17A expression, indicating that TRP channels might promote spinal IL-17A upregulation to facilitate pain. TRP antagonist treatments also attenuated mechanical and thermal hyperalgesia, suggesting that the CIBP detected in this model depends on TRPA1-, TRPV1-, and TRPV4-expressing neurons in DRG. Finally, the authors suggested that the downregulation of TRP channels promoted by LTTL treatment might also contribute to its inhibitory effects on CIBP. In summary, LTTL treatment alleviates bone-cancer-induced nociception by downregulating TRP channel expression in DRG and IL-17A in spinal astrocytes [123].

In another study focusing on CIBP, researchers sought to investigate whether a demethylase named TET1, previously discovered to be overexpressed in the DRG of female rats with bone cancer pain, also contributed to the upregulation of TRPV4 expression. To explore this hypothesis, the authors examined TET1 and TRPV4 expression, along with their cellular localization, in the L4–6 DRG region while assessing pain behaviors after drug administration. The CIBP model was established after the inoculation of the tumor cells in the rats’ tibia and was confirmed by the increased levels of TET1 expression in L4–6 DRG, the development of mechanical allodynia, and the severe destruction of bone structure, with the tumor cells infiltrating the bone marrow cavity. This model further stimulated a gradual increase in TET1 and TRPV4 mRNA levels in L4–6 DRG, which persisted for 21 days after the tumor cells were inoculated. The immunohistochemical analysis proved that TET1 and TRPV4 were co-expressed with markers of small and medium peptidergic neurons—calcitonin gene-related peptide (CGRP) and non-peptidergic neurons—isolectin B4 (IB4). Thus, these co-localization patterns suggest that both of them participated in the nociceptive signaling pathway. In L4–6 DRG, TET1 was found in the cytoplasm and nucleus, and it was presumed to potentially enhance the progression of bone cancer pain through nuclear translocation to modulate gene expression through DNA demethylation processes. The pharmacological inhibition of TET1 resulted in the downregulation of TRPV4 expression in L4–6 DRG, with a subsequent reduction in abnormal pain experienced by rats with CIBP, and the pharmacological blockade of TRPV4 using its selective antagonist HC-067047 led to similar results, as the pain severity was significantly reduced. These findings confirmed that upregulated TET1 and TRPV4 expression in the L4–6 DRG sustained the allodynia in CIBP while a targeted intervention after the inhibition of TET1 or TRPV4 activity may represent a novel therapeutic approach to ameliorating bone-cancer-related pain [125].

A third research group aimed to study the simultaneous expression of extracellular-signal-regulated kinase (ERK1/2) and TRPV4 to obtain an indication of tumor progression in a cancer-induced neuropathy model. They hypothesized that the distant growing tumor continuously activates these two receptors in the afferent sensory neurons in the DRG (L3–5 region) of cancer-induced rats. The tumor cell implantation in the nerve perineurium sheath led to the development of a local non-metastatic tumor, and the model reached its peak six days after cell inoculation, with the significant impairment of thermal and mechanical thresholds. On day 3 after cancer cells’ injection, TRPV4 and ERK1/2 positive immunoreactive (+IR) neurons were overexpressed, with subsequent downregulation from day 7 until day 14 after cancer-induced neuropathy. The increased expression of ERK1/2 is suggestive of tumor progression, and its co-expression with TRPV4 on the same DRG neuronal cells was also confirmed, inferring the mutual synergistic roles of these proteins in pain transduction. The expression profile of TRPV1 and innate immunity toll-like receptor 4 (TLR4) was also assessed in the L3–5 DRG sensory neurons, corroborating the association between these two receptors based on their co-expression on DRG cells of the cancer-induced model. The double-staining of TRPV1 and TLR4 exhibited no alteration in the expression of TRPV1 +IR cells on day 3 after perineural cancer cell inoculation but an increase in TLR4 +IR cells was detected on the same day, followed by a decrease in their expression at 7 and 14 days after tumor injection for both receptors. This prompt upregulation and consecutive decrease in TLR4 +IR cells is related to an immediate immune response and late neural damage due to cancer-induced neuropathy. The diminished expression of TRPV1 and TRPV4 in cancer-bearing animals does not represent a poorly executed function but rather an interconnection between the receptor’s continuous activation by the distant invasive tumor and the increased nociception afterward. Moreover, the TRPV4-selective antagonist (HC-067047) was presented as a successful drug for the treatment or relapse of several neuropathies [105,113,116]. In this cancer-neuropathy model, the inhibition of TRPV1 and TRPV4 channel activity by their selective antagonists, capsazepine and HC-067047, respectively, transiently reversed the cancer-induced thermal and mechanical allodynia in a dose-dependent manner. Finally, ruthenium red, a calcium signaling inhibitor, was effective for thermal hyperalgesia in different periods but reversed the mechanical allodynia only at the higher tested dose, suggesting a preferential antagonism for TRPV1 over TRPV4 [104].

Furthermore, PAR-2 and TRPV4 were found to be co-expressed in DRG neurons, suggesting that the PAR-2/TRPV4 pathway in primary afferent fibers is essential in nociception [81,122,127]. Trypsin, a protease known to sensitize primary afferent fibers through PAR-2 activation, is involved in several cancers, including oral cancer [127,128]. Thus, the last group intended to clarify whether the trypsin/PAR-2/TRPV4 axis mediates cancer pain through changes in trigeminal ganglion (TG) neurons projecting to the tongue in an orofacial cancer pain model. The characteristic sign observed after squamous cell carcinoma (SCC) implantation was the progressive cell invasion and development of a large tumor mass at the inoculation site, causing head and tongue nociception from the second day after cancer induction onwards. Trypsin overexpression in the tongue after SCC inoculation was correlated with mechanical sensitization of the tongue due to PAR-2 sensitization in TG primary afferent fibers caused by tumor-derived trypsin. To confirm that trypsin/PAR-2 signaling is involved in the mechanical allodynia, a trypsin inhibitor protein (STI) and PAR-2 antagonist (FSLLRY-NH_2_) were also administered to the tongue, ameliorating the nociceptive symptoms triggered by tumor cells’ implantation. However, this inhibition was incomplete, suggesting that several proteases from the SCC may be involved in nociception. The hypothesis of the overexpression of PAR-2 and TRPV4 in TGs after SCC inoculation to the tongue was also investigated but was not confirmed in this model, as the amounts or expression patterns for both proteins remained unchanged. Following the premise that PAR-2 signaling modulates TRPV4 activity through phosphorylation and TRPV4 may be phosphorylated without altering the total amount of protein [127], the next step involved quantifying TRPV4-phosphorylated protein levels in the TGs of SCC-inoculated rats. The upregulation of the TRPV4 protein was prominent but markedly prevented by the administration of a PAR-2 antagonist to the tongue, establishing the functional relationship between PAR-2 and TRPV4 in orofacial cancer pain. However, detecting intracellular signaling between PAR-2 and TRPV4 induced by trypsin was not possible. Moreover, the hindered nociceptive behavior evoked by the SCC model was reversed by PAR-2 or TRPV4 antagonism (RN-1734) in a dose-dependent manner. The outcomes from this study corroborate that trypsin resulting from tongue SCC activates PAR-2, followed by TRPV4 phosphorylation in primary afferent neurons and sensitization on the tongue, ending with nociception enhancement via the potentiation of neuronal activity in the TG [82].

Although preliminary, these findings have brought into view different approaches regarding TRPV4’s involvement in cancer pain (Figure 3).

The four studies were performed in rats (male or female) and investigated the role of TRPV4 channels solely in mechanical nociception. As described, only three types of cancer pain (bone, perineural, and orofacial) models investigated to date have covered the description of multiple mechanisms sensitizing TRPV4 (interleukins, DNA demethylation, ERK1/2, and PAR-2). Selective (HC-067047 and RN-1734) and non-selective (gadolinium and ruthenium red) antagonists were also employed, and the overexpression of these channels was correlated with increased pain perception. In addition to research in more varied cancer pain models, an interesting approach for further studies would be to compare whether TRPV4’s amount or expression patterns are similarly distributed in males and females. Moreover, investigating these cancer pain models in mice would provide the advantage of using animals carrying genetic deletion for the *Trpv4* gene. These initial findings may contribute to discovering and developing new therapeutic targets to alleviate the pain caused by cancers.

### 3.4. TRPV4 Antagonists as Therapeutic Drugs

As described so far, TRPV4 is involved in sensory responses triggered by an array of endogenous and exogenous stimuli. This ion channel is constitutively expressed and capable of maintaining spontaneous activity without agonist stimulation, suggesting a fundamental role in physiological functions and making it an attractive therapeutic target for treating multiple pathologies [88,122]. In animal models, TRPV4 antagonists such as HC-067047, RN-1734, and GSK2193874 have therapeutic effects on many disorders, including cardiovascular and ocular diseases, osteoarthritis, and bladder hyperactivity. These antagonists have also been reported to decrease nociception in preclinical models of inflammatory and neuropathic pain [87,88,90,122]. However, clinical evidence of the benefit of TRPV4 in pain control remains lacking.

Research efforts have been made in the discovery and development of TRPV4 antagonists as medicines, and many patent publications on TRPV4 antagonists describe its therapeutic applications in a multitude of pathologies, including pain, glioma, spinal cord injury, cerebral edema, cough, heart failure and ischemic heart disease, and dermatological conditions, among others [88]. However, it is worth noting that the development of TRPV4 antagonists as a potential therapeutic intervention is in the early stages. So far, the TRPV4 antagonist, referred to as GSK2798745, is the only one being considered as a clinical candidate [89]. This compound exhibited promising pharmacokinetics properties and a favorable safety profile in clinical studies, qualifying it as a feasible option for further investigation. In the most recent Phase I clinical trial, the oral administration of GSK2798745, whether in liquid form (where the drug powder was dissolved in water) or capsule formulation, was well-tolerated by healthy volunteers and stable heart failure patients. with no adverse effects reported [89].

As a result, GSK2798745 was selected for further evaluation in long-term clinical trials not only in heart failure but also in other relevant medical conditions. Currently, TRPV4 antagonists are being investigated in clinical trials to determine the efficacy of these compounds in the treatment of patients with pulmonary edema (NCT02135861) and congestive heart failure (NCT02497937) [88,117]. Although there is still no perspective on the testing of TRPV4 antagonists for pain management, research efforts in this direction may reveal new relevant information that can be used to formulate novel drug therapies targeting pain-related conditions. Altogether, the above investigations are critical to understanding the potential of TRPV4 antagonists across many clinical contexts and provide hope for the development of new treatment strategies to address unmet medical needs.

The main exogenous agonists of TRPV4 comprise various compounds, ranging from ω-3 polyunsaturated fatty acids to plant-derived molecules and synthetic agents, as mentioned before, all of which have shown potential but still are under-explored as antinociceptive agents in the clinical setting. While the investigation of these agonists remains pending in a clinical context, the need to verify if a TRPV4 channel agonist could cause the defunctionalization of nociceptors, as observed with the capsaicin mechanism of action, continues to exist. To date, only in vitro experiments have demonstrated that TRPV4 activation by agonists such as PAR-2 and 4α-PDD sensitized sensory afferents to mechanical stimulation and that the agonist GSK10116790A caused the activation, followed by the desensitization, of TRPV4 channels [60,129]. Regarding this, the application of TRPV4 antagonists remains the predominant approach for eliciting antinociceptive effects, but again, in the current scenario, it was solely tested in animal models.

Due to the vast array of physiological functions performed by TRPV4 in numerous body systems in health and diseases, it is crucial to assess the potential safety concerns associated with the therapeutic use of TRPV4 agonists and antagonists. Specifically, while inhibiting the pro-inflammatory actions of TRPV4 could have therapeutic value in certain circumstances, its broad-spectrum inhibition could lead to detrimental effects. For instance, interfering with the endogenous function of TRPV4 in the early stages of sepsis could be dangerous, as TRPV4 is necessary for host responses to mycobacteria in the early phases of infection [130]. Collectively inhibiting the receptor may also cause more adverse effects by compromising the immune response, which could occur through mechanisms such as declining the infiltration of neutrophils along with restraining macrophage migration, thus affecting the body’s ability to defend against pathogens. Additionally, lengthening or blunting TRPV4 agonists may limit the generation of oxidative stress, a closely related component to immune function [67,68,90,122,129]. Therefore, developing TRPV4 antagonists with therapeutic indications represents a significant challenge. The process requires identifying molecules that are specific enough to target only the intended TRPV4 channels while sparing those involved in essential physiological processes. This selectivity allows for the development of drugs with no harmful adverse effects and ensures the safe and beneficial use of TRPV-targeted pharmacological intervention in several morbidities [90,122].

In summary, the research on TRPs has blossomed over the last decade as these channels show potential as novel therapeutical targets that could be useful in treating several conditions, including inflammatory and neuropathic pain, cancer pain, and migraine. Although numerous TRP channel drugs have entered clinical trials, only agonists of TRPA1 (eugenol), TRPV1 (resiniferatoxin and capsaicin), and TRPM8 (menthol) have been approved for medical use by the US Food and Drug Administration (FDA) [117].

## 4. Conclusions

As previously outlined, the recent advances in cancer therapy have improved patients’ life expectancy, suggesting that more individuals are likely to endure long-lasting symptoms arising or persisting beyond the conclusion of the treatment. Cancer pain represents the most threatening consequence emerging from cancer therapy or cancer itself, and the mechanisms related to cancer pain remain unclear. Multiple molecules and complex processes are related to pain generation and progression, for instance, CIBP, a debilitating symptom and the most frequent cause of cancer-related pain, and CIPN, an adverse effect arising from the antitumor agents’ neurotoxicity. The interactions between the tumor, primary afferent nociceptor, and immune system contribute to the unpredictable aspects of cancer, turning pain management into a challenge. Despite their mixed pathophysiology, opioids still stand as a cornerstone of pharmacological therapies for moderate to severe cancer pain, despite their critical adverse effects.

Evidence suggests that the activation of different ion channels is involved in cancer growth and metastasis and implicated in the initiation and maintenance of cancer pain. As TRPV4 modulates several processes engaged in physiological and pathological responses, different mechanisms might sensitize TRPV4 channels as well. Extensively expressed and found in primary afferents neurons and immune and skin cells, these receptors are crucial for the maintenance of painful peripheral neuropathies. Their ability to respond to an array of stimulus types highlights TRP channels as attractive therapeutic targets and promising options for treating cancer-related pain, particularly in cases where conventional therapeutic approaches demonstrate limited effectiveness.

Considering TRPV4’s wide distribution and involvement in numerous physiological processes, multiple mechanisms may contribute to its involvement in pain relief. In cancer-related pain, TRPV4 channels are directly activated due to the excessive generation of oxidative stress by-products (e.g., H_2_O_2_), acidification, and the release of mediators such as TNF-α by the tumor microenvironment. The resulting process includes peripheral sensitization and the subsequent development of neuropathic pain. In this context, it is possible to assume that the selective modulation of TRPV4 channel activity might be beneficial. Furthermore, the channel activation also induces the release of pro-inflammatory mediators and neurotransmitters from sensory neurons, and the neuronal excitability of pain-sensing neurons increases. Modulating these processes could alleviate the inflammatory response and normalize the neurotransmitter’s release, as well as the neuron excitability, possibly suppressing nociceptive signals and thus reducing pain transmission and perception.

Preclinical models have described the participation of TRPV4 in nociception and analgesia over CIPN. These studies underlined TRPV4 channel modulators as a promising approach for alleviating CIPN symptoms. However, all the analyses used chemotherapy drugs administered to healthy animals. A more realistic strategy would be to provide antineoplastic treatment after the inoculation of tumor cells to induce a proper cancer model. Moreover, a limited number of studies have investigated the TRPV4 channel’s involvement in cancer-induced pain. Substances such as menthol, a TRPM8 agonist, and capsaicin and resiniferatoxin, both TRPV1 agonists, have been widely used and investigated in clinical studies. Clinical expertise regarding TRPV4 antagonists is restricted to a sole agent, GSK2798745, tested in a small number of healthy volunteers or individuals suffering from cardiac and respiratory disorders. Despite TRPV4 antagonists reaching clinical development, none have been approved for clinical practice yet, revealing the importance of better understanding the physiological role played by these ion channels in tissues and organs, in addition to converging the efforts to develop preclinical models with a higher affinity in clinical translation. Finally, TRPV4 is co-expressed with different TRP channels (TRPV1 and TRPA1), and the blockage of these receptors may also reduce cancer-induced nociception. However, we consider that research on the sole function of this particular channel in cancer pain is still incipient.

## Figures and Tables

**Figure 1 cancers-16-01703-f001:**
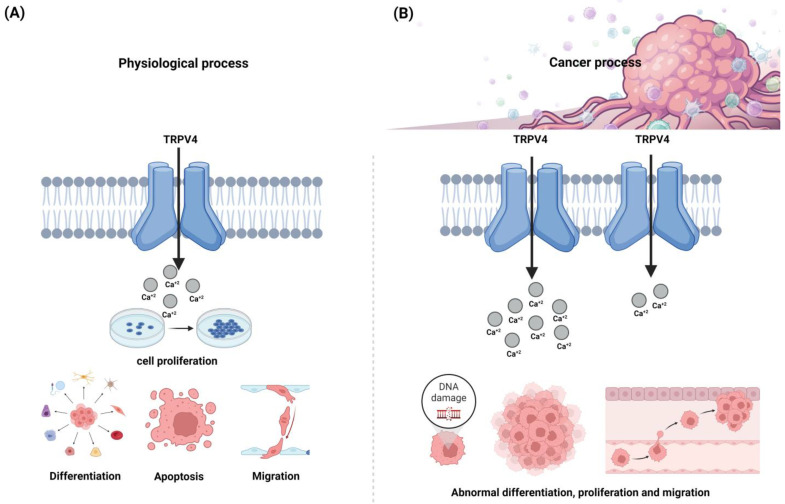
Transient receptor potential vanilloid 4’s (TRPV4’s) involvement in cancer etiology. TRPV4 is a non-selective calcium channel widely expressed in many mammalian cells and tissues, including the immune, sensory, and central nervous system cells. It mediates calcium influx and promotes the transduction of nociceptive stimuli. (**A**) TRPV4 channels are associated with several physiological processes and functionally related to cell proliferation, differentiation, apoptosis, and migration, among other mechanisms. (**B**) The abnormal expression of TRPV4 and the altered ion transport facilitated by this channel varies across different tumors and contributes to atypical cell proliferation and prolonged survival, leading to the uncontrolled tumor growth and differentiation patterns observed in malignant cells. Created with BioRender.com.

**Figure 2 cancers-16-01703-f002:**
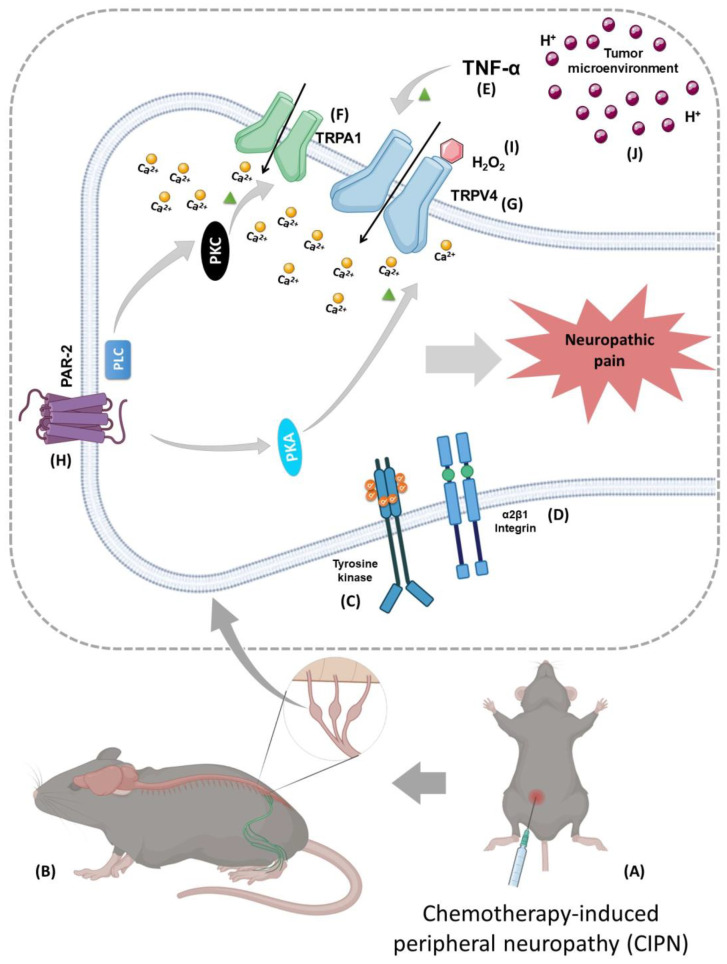
Transient receptor potential vanilloid 4 (TRPV4) involvement in the mechanism of neuropathic pain, a typical symptom described in chemotherapy-induced peripheral neuropathy (CIPN) etiology. Preclinical models showed that (**A**) after antineoplastic drug injection, CIPN stems from the activation of TRPV4 channels in the (**B**) sensory nociceptive neurons located in the dorsal root ganglia (DRG). Some mechanisms involved in CIPN after TRPV4 activation are as follows: (**C**,**D**) tyrosine kinase receptor phosphorylation leads to α2β1 integrin activation; (**E**–**G**) chemotherapy agents increase TNF-α expression, leading to TRPA1 and TRPV4 upregulation in DRG neurons and (**H**) triggering the activation of protease activating receptor 2 (PAR-2) by tryptase, followed by the activation of protein kinases A (PKA) and C (PKC), which leads to TRPV4 channel sensitization; (**I**) TRPV4’s direct activation via the excessive production of hydrogen peroxide (H_2_O_2_) causes calcium influx, which induces neuropathic pain; (**J**) acidification and mediators from the tumor microenvironment modulate TRPV4 channels, resulting in peripheral sensitization. The ▲ (green triangles) represent the channel’s activation; ⬢ (salmon hexagon) represents hydrogen peroxide (H_2_O_2_); ⬤ (purple circles) represent protons (H^+^), and ⬤ (yellow circles) represent calcium ions.

**Figure 3 cancers-16-01703-f003:**
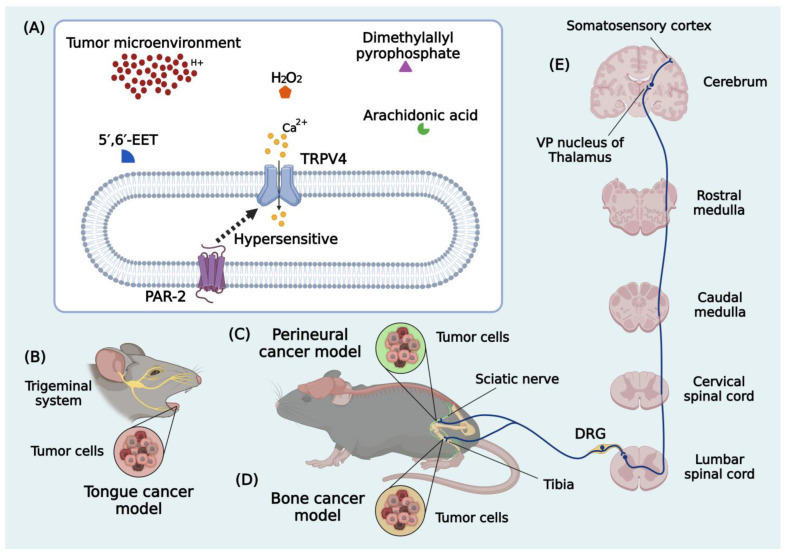
Transient receptor potential vanilloid 4 (TRPV4) involvement in preclinical cancer pain models. The activation of TRPV4 by (**A**) endogenous agonists (osmolarity, pH, H_2_O_2_, PAR-2, dimethylallyl pyrophosphate, arachidonic acid, and 5′,6′-EET) promotes Ca^2+^ influx. (**B**) Orofacial cancer activates nociception via the trigeminal ganglion system. (**C**) Perineural cancer and (**D**) bone cancer trigger (**E**) peripheral afferent neurons to cause nociception in different cancer pain models. Abbreviations: H_2_O_2_, hydrogen peroxide; 5′,6′-EET, 5′,6′-epoxyeicosatrienoic acid; PAR-2, protease-activated receptor; Ca^2+^, calcium ions; DRG, dorsal root ganglion; VP, ventral posterior. Created with BioRender.com.

**Table 1 cancers-16-01703-t001:** Transient receptor potential vanilloid 4 (TRPV4) involvement in preclinical cancer pain models.

Rodent Type	Cells and Site of Inoculation	Model	Dose, Time of Treatment, and Route of Administration	Ref.
**Female** **Sprague-Dawley (SD) rats**	Walker 256 tumor cells; bone marrow	Bone cancer model	- Gadolinium (a non-selective TRPV4 antagonist, 10 µg, intrathecal); - The treatment was provided 17–20 days post tumor cells’ inoculation.	[123]
**Female** **Sprague-Dawley (SD) rats**	Walker 256 tumor cells; bone marrow	Bone cancer model	- HC067047 (a selective TRPV4 antagonist, 0.1 mg/kg, 20 µL; intrathecal); - Treatment 7 days post tumor cells’ inoculation.	[125]
**Male** **Copenhagen rats**	AT-1 cells; sciatic nerve	Perineural cancer model	- HC-067047 (a selective TPV4 antagonist, 10 and 20 mg/kg; s.c.); - The treatment was carried out on day 7 post tumor cells’ inoculation.	[104]
**Male** **Fischer** **rats**	SCC-158 cells; tongue	Orofacial cancer model	- RN-1734 (a selective TRPV4 antagonist, 6 µL, administered to the tongue); - The treatment was carried out on day 7 post tumor cells’ inoculation.	[82]

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
