# Peer review of "Targeting TRPV4 Channels for Cancer Pain Relief"

_cancers, 2024, doi:10.3390/cancers16091703_

Round 1

Reviewer 1 Report

Comments and Suggestions for Authors

 I have a few queries:

1. Is it advisable to use the full form of all abbreviations when they are first introduced? For instance, on line 87, the author uses the abbreviation TRPM8.

2. The author discusses different receptors and pathophysiological processes related to cancer pain. Therefore, what is the significance of solely modulating the TRPV4 receptor in managing cancer-related pain?

3. The author outlines various mechanisms involved in pain perception through TRPV4 channels. By modulating the channel, which mechanisms may contribute to pain relief?

4. TRPV4 channels are functionally linked to cell proliferation, differentiation, apoptosis, migration, etc. Apoptosis is utilized to eliminate irreparably damaged cells from the body. Therefore, could modulating this receptor impact apoptosis and potentially increase cancer progression in patients?

Reviewer 2 Report

Comments and Suggestions for Authors

The paper is well written and interesting and offers an extensive summary of TRPV4 action in modulating cancer pain, arising from bone metastasis as well as from CIPN.

It covers an interesting aspect about the role of TRPV4 on cancer development, and offers an  insight  about potential therapeutic use of TRPV4 agonist.

To add more clarity, could the author provide a table or a figure about the action of TRPV4 on cancer progression, summarising the evidence?

Also a table and/or a  figure that better specify the mechanism of action of TRPV4 in developing cancer pain or CIPN is suggested, to improve reader's comprehension of the text. Could the authors include some mechanisms such as those in the text?

Finally, the authors proposed a role for TRPV4 antagonist to treat cancer pain, but  they no specify in which form (topical one such as capsaicin?) And what about agonist of TRPV4 (like capsaicin one again)? Could they have a role in cancer pain treatment?What about potential side effects?

Comments on the Quality of English Language

The paper is clear and well written.
